# High-temperature flexible WSe$_2$ photodetectors with ultrahigh photoresponsivity

Yixuan Zou[1,2,3], Zekun Zhang[1,2,3], Jiawen Yan[1,2,3], Linhan Lin[1], Guanyao Huang[4], Yidong Tan[1] ✉, Zheng You[1,2,3] ✉ & Peng Li [1,2,3] ✉

The development of high-temperature photodetectors can be beneficial for numerous applications, such as aerospace engineering, military defence and harsh-environments robotics. However, current high-temperature photodetectors are characterized by low photoresponsivity (<10 A/W) due to the poor optical sensitivity of commonly used heat-resistant materials. Here, we report the realization of h-BN-encapsulated graphite/WSe2 photodetectors which can endure temperatures up to 700 °C in air (1000 °C in vacuum) and exhibit unconventional negative photoconductivity (NPC) at high temperatures. Operated in NPC mode, the devices show a photoresponsivity up to $2.2 \times 10^6$ A/W, which is ~5 orders of magnitude higher than that of state-of-the-art high-temperature photodetectors. Furthermore, our devices demonstrate good flexibility, making it highly adaptive to various shaped surfaces. Our approach can be extended to other 2D materials and may stimulate further developments of 2D optoelectronic devices operating in harsh environments.

Photodetectors are considered to be the core of modern communication components. As the development of aerospace, military, underground exploration, and harsh environment robotics, photodetector capable of operating at harsh environments are highly desired. Current high-temperature photodetectors are mainly based on wide-bandgap materials, such as SiC, GaN, or Ga$_2$O$_3$. Their optical sensing capability is barely satisfactory. A typical SiC photodetector can endure 550 °C and its photoresponsivity (one of the most essential figures of merit for photodetector) is only 0.54 A/W[1]. GaN photodetector is able to operate in a wide temperature range from −196 °C to 527 °C[2], but it demonstrates very low photoresponsivity of 0.02 A/W. So et al. reported a AlGaN/GaN photodetector with operation temperature of 200 °C and photoresponsivity of 5 A/W[3]. Zhou et al. developed a Ga$_2$O$_3$ photodetector with operation temperature of 200 °C and photoresponsivity of 0.1 A/W[4]. There is a trend that photodetectors which can endure high temperature demonstrate very low photoresponsivity. The photoresponsivity of state-of-the-art high-temperature photodetectors is usually below 10 A/W. As such, weak

light detection at high temperature is challenging. Furthermore, flexibility is an important development trend for next generation photodetectors. Flexible high-temperature photodetectors are desired in many applications, such as aeroengine blade with flexible (optical) sensors attached on its curved surface, and fully adaptive soft robot working in high-temperature environment with highly flexible sensors integrated on soft artificial muscle. However, high-temperature photodetectors reported so far are rigid. Limited by the poor thermal stability of traditional flexible materials[5] (polyethylene terephthalate (PET), polyimide (PI), polydimethylsiloxane (PDMS) and so forth), the temperatures that existing flexible optoelectronic devices can endure are below 300 °C[6,7]. Consequently, highly-sensitive high-temperature photodetectors with good flexibility are highly desired.

Two-dimensional transition metal dichalcogenides (TMDs) have drawn a great deal of attention in optoelectronics field due to their fascinating physical properties[8,9] and facile preparation. They can reach monolayer or few layer by mechanical exfoliation[10], liquid-phase exfoliation[11], or chemical vapor deposition method[12,13]. Although

[1]State Key Laboratory of Precision Measurement Technology and Instruments, Department of Precision Instruments, Tsinghua University, Beijing 100084, China. [2]Key Laboratory of Smart Microsystem (Tsinghua University) Ministry of Education, Beijing 100084, China. [3]Beijing Advanced Innovation Center for Integrated Circuits, Beijing 100084, China. [4]Key Laboratory for Thermal Science and Power Engineering of Ministry of Education, Beijing Key Laboratory of CO2 Utilization and Reduction Technology, Department of Energy and Power Engineering, Tsinghua University, Beijing 100084, China. ✉e-mail: Tanyd@mail.tsinghua.edu.cn; yz-dpi@mail.tsinghua.edu.cn; pengli@mail.tsinghua.edu.cn

graphene is the most well-investigated 2D material, TMDs have the advantages of higher sensitivity and lower dark current, and demonstrate remarkable optical sensing capability at room temperature. The photoresponsivity of monolayer $MoS_2$ photodetector reaches 880 A/W[14]. Xie et al. reported photodetectors based on multilayer $WSe_2$ flake with photoresponsivity of $1.5 \times 10^5$ A/W[10]. Monolayer $WSe_2$ photodetector demonstrates higher photoresponsivity of $1.8 \times 10^5$ A/W[12]. Additionally, good mechanical properties (large fracture strain[15,16]) make TMDs promising channel materials for flexible photodetector. However, TMD electronic/optoelectronic devices cannot survive at high-temperature. $MoS_2$ starts to oxidize and degrade at 300 °C in air, resulting in many triangular etch pits on its surface[17–19]. The edge of $WSe_2$ flake begins to oxidize at 300 °C in air[20]. It hinders the application of TMDs in aerospace and many other harsh environments. Importantly, it hinders the exploration of optoelectronic properties of TMDs at ultrahigh temperature.

Here, we developed $WSe_2$ photodetectors with h-BN (hexagonal boron nitride)/GF (graphite flake) heterostructure protection which can endure 700 °C in air and 1000 °C in vacuum. Unconventional negative photoconductivity (NPC) phenomenon appeared at high temperature. Operated in NPC mode, the device exhibited photoresponsivity of $2.2 \times 10^6$ A/W, which is ~$10^5$-fold higher than that of state-of-the-art high-temperature photodetectors, and even higher than that of existing $WSe_2$ photodetectors. The photodetector demonstrated good flexibility and realized in situ high-temperature optical sensing under bending state.

## Results

### WSe2 photodetector fabrication and characterization

We fabricated $WSe_2$ field-effect transistor (FET) as photodetector with h-BN encapsulation and GF source/drain electrodes (Fig. 1a) by mechanically stacking each atomic layer sequentially on freshly cleaved mica surface (Supplementary Fig. 1) followed by photolithography, Pt deposition, and lift-off for top gate fabrication. The optical microscope image of a representative $WSe_2$ device is shown in Fig. 1c (Pt top gate is not shown). Mica substrate with thickness of ~100 μm is highly flexible (bending radius <2 mm, Fig. 1b), high-temperature-resistant, and transparent. Additionally, compared with other flexible substrates, mica provides atomically flat terraces over large areas, so 2D materials can approach the limit of atomic flatness on mica surface and get rid of the microscopic corrugations which result in carrier scattering and degradation of electrical properties[21]. Multilayer $WSe_2$ flake was chosen as channel material of high-performance

photodetector. Hexagonal boron nitride (h-BN) has been reported in previous work to be excellent oxygen-resistant coating[22]. We placed two h-BN flakes with thickness of ~50 nm on top and bottom of $WSe_2$, respectively as high-temperature encapsulation, where top h-BN also serves as gate dielectric (Supplementary Fig. 2). The 2D material flakes were all characterized by Raman spectroscopy (Supplementary Fig. 3 and Fig. 2c). Sharp Raman peaks imply that the 2D materials studied in this work have nearly perfect lattice structures. Figure 1d is the high-resolution transmission electron microscopy (HRTEM) cross-sectional image of h-BN/GF/$WSe_2$/h-BN van der Waals heterostructures in the $WSe_2$ FET (Supplementary Fig. 4). The thickness of monolayer $WSe_2$ (~0.7 nm) is consistent with reported values[23,24]. It can be seen clearly that the 2D materials are atomically flat and no air gap is observed at interface, indicating good encapsulation, which is essential for high-temperature protection.

### High-temperature-resistant capability

To investigate the high-temperature-resistant capability of our device, the $WSe_2$ FET was heated at 500 °C, 600 °C, and 700 °C for 15 min sequentially in an open quartz furnace. The high-temperature experiments in this work were all carried out in air, unless otherwise noted. $WSe_2$ flake protected by h-BN showed negligible change after heating (Fig. 2b, Pt top gate is not shown). Raman spectra showed $E^1_{2g}$ peak (~249 cm$^{-1}$) and $A_{1g}$ peak (~258 cm$^{-1}$) corresponding to $WSe_2$, whereas no $WO_3$ character peaks (~700 cm$^{-1}$ or ~810 cm$^{-1}$) was observed[25–27] (Fig. 2c), implying that $WSe_2$ was not oxidized. After 700 °C heating, flexible mica substrate was in good shape as well. Then the same device was heated at 750 °C for 15 min. The $WSe_2$ channel still showed negligible change (Supplementary Fig. 5), but mica substrate became brittle and less transparent. As control groups, bare $WSe_2$ and $WSe_2$ covered with 100 nm $Al_2O_3$ (a widely used oxidation-resistant coating) by atomic layer deposition were heated at 500 °C for 15 min. In both situations, the $WSe_2$ flakes were strongly oxidized and became almost transparent (Supplementary Fig. 6) inasmuch as $WO_3$ is transparent under visible light. Raman spectrum shows prominent $WO_3$ character peaks around 700 cm$^{-1}$ and 810 cm$^{-1}$ (Fig. 2c). Therefore, bare $WSe_2$ has poor thermal stability, which is consistent with previous reports[20]. The h-BN is superior to $Al_2O_3$ as high-temperature protection layer.

Next, we investigated the high-temperature-resistant capability in vacuum of our devices. Since mica substrate cannot endure temperature above 750 °C, we replaced it with Si substrate with 300 nm $SiO_2$ on top. We heated the $WSe_2$ FETs at 1000 °C for 15 min in vacuum with Argon flow rate of 100 sccm.

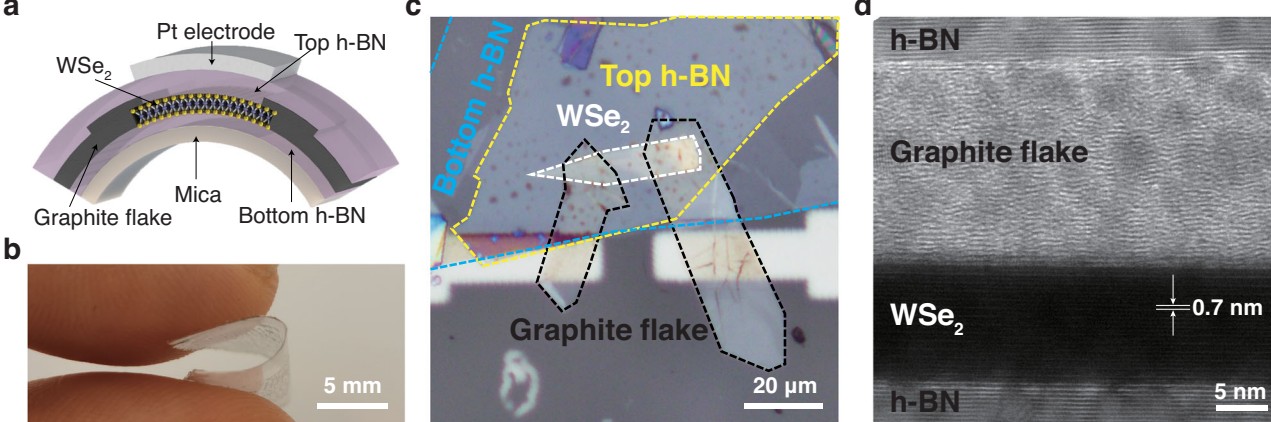

**Fig. 1 | High-temperature-resistant flexible WSe2 photodetector. a** Schematic view of $WSe_2$ photodetector with h-BN encapsulation and GF (graphite flake) electrodes on mica substrate. **b** Optical image of the $WSe_2$ device on flexible mica substrate. **c** Optical microscopy image of the $WSe_2$ device which is consist of 5 pieces of 2D-material flakes (Pt top gate is not shown). Dashed lines are used to indicate the outline of $WSe_2$ (white), GF (black), bottom h-BN (blue) and top h-BN (yellow). **d** HRTEM cross-sectional image of h-BN/GF/$WSe_2$/h-BN van der Waals heterostructure in $WSe_2$ device.

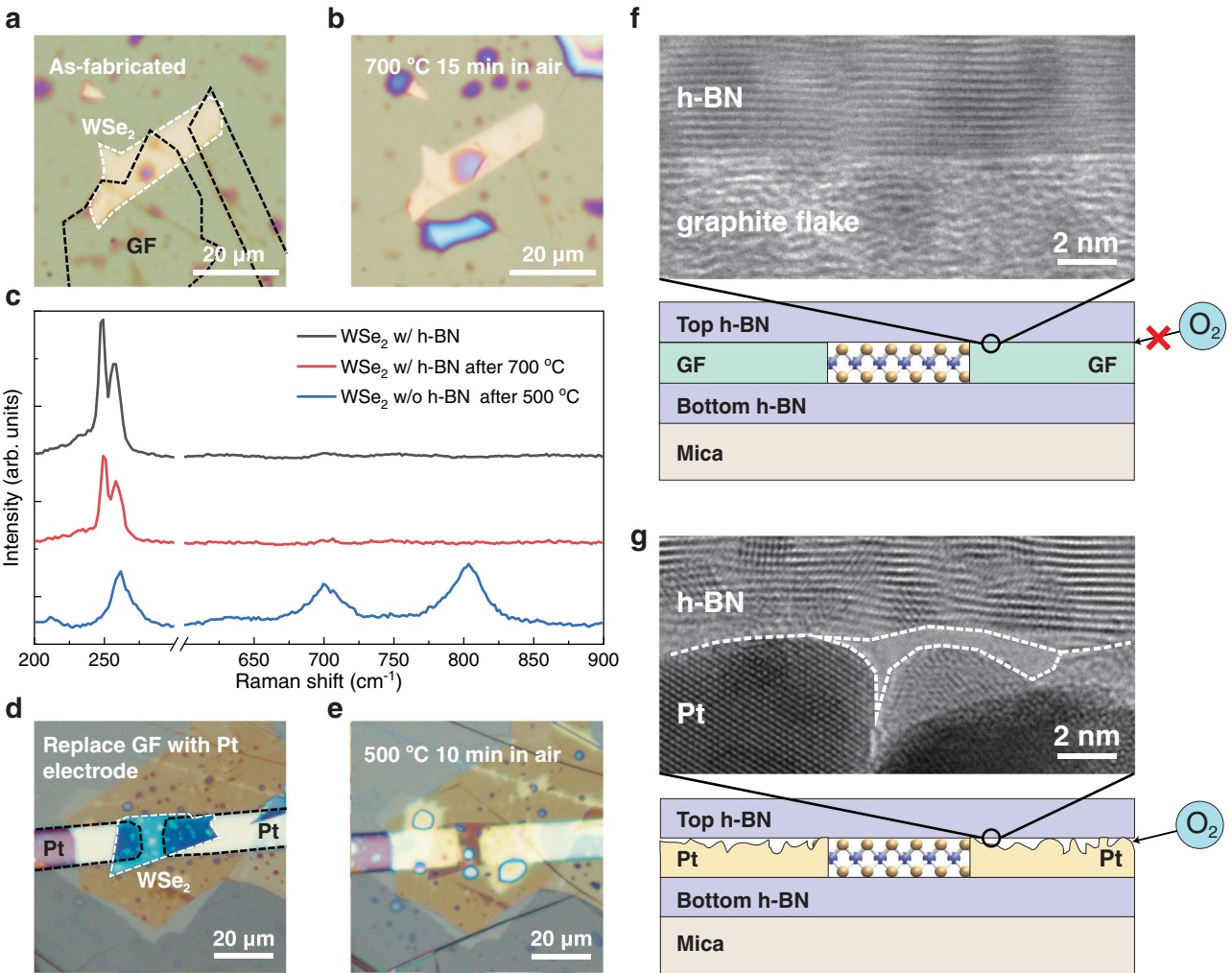

**Fig. 2 | High-temperature-resistant capability. a** As-fabricated WSe$_2$ FET. Dashed lines are used to indicate the outline of WSe$_2$ (white) and GF (black). **b** WSe$_2$ FET after 700 °C heating. **c** Raman spectra of WSe$_2$ with h-BN encapsulation and GF electrodes before (black) and after (red) 700 °C heating. Raman spectrum of a bare WSe$_2$ after 500 °C heating (blue). **d** As-fabricated WSe$_2$ FET with h-BN encapsulation and Pt electrodes. **e** WSe$_2$ FET with Pt electrodes after 500 °C heating. Dashed lines are used to indicate the outline of WSe$_2$ (white) and Pt electrodes (black).

**f** HRTEM cross-sectional image of h-BN/GF interface and schematical cross-sectional view of WSe$_2$ FET with GF electrodes. Oxygen cannot diffuse into h-BN encapsulation through h-BN/GF interface. **g** HRTEM cross-sectional image of h-BN/Pt interface and schematical cross-sectional view of WSe$_2$ FET with Pt electrodes. A white dashed line is used to indicate the bottom interface outline of h-BN and top interface outline of Pt electrodes. Oxygen can diffuse into h-BN encapsulation through h-BN/Pt interface.

The WSe$_2$ FET showed negligible change after heating (Supplementary Fig. 7). Raman spectrum demonstrates prominent WSe$_2$ character peaks and no WO$_3$ character peak, indicating that WSe$_2$ lattice structure remained intact after heating. As a control group, bare WSe$_2$ flake on SiO$_2$/Si substrate completely vanished after 1000 °C heating (Supplementary Fig. 7). Therefore, the h-BN/GF structure effectively protects WSe$_2$ at ultrahigh temperature of 1000 °C in vacuum. Dark current increased slightly after 700 °C in air and 1000 °C in vacuum annealing (Supplementary Fig. 8). The temperature that our devices can tolerate is much higher than that of current 2D material devices, both in air and vacuum environments (Table 1). Our research expands the working temperature range of 2D materials, allowing the good electrical properties of 2D materials to be applied in high-temperature environments.

Graphite flake electrodes are essential to high-temperature protection. We replaced the GF in WSe$_2$ FET with conventional (Pt) metal electrodes (Fig. 2d). After heating at 500 °C for 10 min, WSe$_2$ flake within different sizes of h-BN encapsulation was strongly oxidized (Fig. 2e, Supplementary Fig. 9 and Supplementary

Table 1). This is because Pt film deposited by sputtering has much larger surface roughness than that of GF. Top h-BN cannot completely conform to the topography of Pt surface (HRTEM cross-sectional image in Fig. 2g), leading to oxygen molecules diffusion into h-BN encapsulation through Pt/h-BN interface. Atomic force microscopy (AFM) characterization demonstrates that the height variation of Pt surface is ~6.8 nm (Supplementary Fig. 10), while the height variation of GF measured (~0.6 nm) appears to be limited by instrument noise and is identical to that obtained from the surface of highly oriented pyrolytic graphite (HOPG) which approaches the limit of atomic flatness. As such, good contact is formed between h-BN and GF (HRTEM cross-sectional image in Fig. 2f) which can effectively prevent oxygen diffusion. To further prove the oxygen diffusion through Pt/h-BN interface, we transferred a WSe$_2$ flake on Pt surface and covered it with h-BN. After heating at 500 °C for 15 min, WSe$_2$ inside h-BN/Pt encapsulation was strongly oxidized (Supplementary Fig. 11). GF/WSe$_2$/h-BN sandwich structure was prepared and tested under the same experimental condition. The WSe$_2$ flake inside h-BN/GF encapsulation was still in good shape after heating. Therefore, GF electrode prevents

**Table 1 | Comparison of temperature tolerance in different 2D material devices**

| Device | Device Type | Temperature Tolerance | Environment | Flexibility | Ref |
|---|---|---|---|---|---|
| $MoS_2$ | Photodetector | 300 °C | in air | flexible | [36] |
| $MoS_2$ | FET | 125 °C | in air | rigid | [37] |
| $MoS_2$ | Photodetector | 200 °C | in air | rigid | [38] |
| $MoS_2$ | Synaptic Transistor | 350 °C | in air | rigid | [44] |
| $MoS_2$ | TFT | 107 °C | in air | rigid | [45] |
| $MoS_2$ | FET | 107 °C | in air | rigid | [46] |
| $MoS_2$ | TFT | 223 °C | in air | rigid | [47] |
| $ReS_2$ | FET | 102 °C | in air | rigid | [48] |
| MoTe2 | FET | 400 °C | in $N_2$ | rigid | [49] |
| Graphene | FET | 500 °C | in vacuum | rigid | [50] |
| Graphene | Sensor | 650 °C | in vacuum | flexible | [51] |
| $WSe_2$ | Photodetector | 700 °C | in air | flexible | This work |
| $WSe_2$ | Photodetector | 1000 °C | in vacuum | flexible | This work |

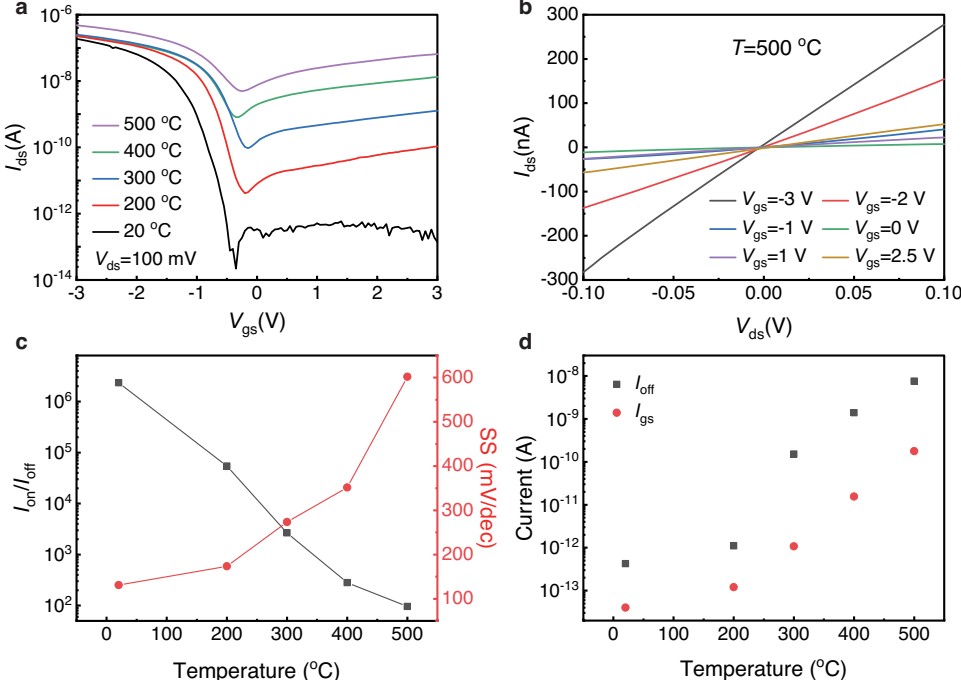

**Fig. 3 | Electrical characteristics of WSe₂ FET measured under high temperature. a** Transfer curves of WSe$_2$ FET measured under different temperatures. $I_{ds}$ is the drain current and $V_{ds}$ is voltage between drain and source. **b** $I_{ds}$-$V_{ds}$ curves of WSe$_2$ FET measured under 500 °C. $V_{gs}$ is voltage between gate and source. **c** $I_{on}/I_{off}$ ratio and SS (subthreshold swing) of WSe$_2$ FET derived under different temperatures. $I_{on}$ is the on-state current and $I_{off}$ is the off-state current. **d** $I_{off}$ and $I_{gs}$ (h-BN leakage current) measured under different temperatures.

oxygen diffusion and plays an important role in high-temperature protection.

## High-temperature electrical properties

We systematically investigated the temperature-dependent electrical properties of the WSe$_2$ FET in dark environment without the influence of photoexcitation. Electrical measurements were carried out below 550 °C for safety reason. Figure 3a demonstrates the transfer curves of a representative WSe$_2$ device with channel width/length of 10 μm/5 μm (source/drain voltage $V_{ds}$ = 100 mV). At room temperature (20 °C), on/off ratio of $2 \times 10^6$ and subthreshold swing (SS) of 130 mV/dec were obtained (SS = d$V_{gs}$/dlg$I_{ds}$, where $V_{gs}$ is gate bias, and $I_{ds}$ is source/drain current). Carrier mobility derived from p branch reached ~35 cm$^2$/V·s. The excellent room-

temperature electrical performance is comparable to that of the best WSe$_2$ field-effect transistors reported[28,29]. As the temperature increased, larger $I_{ds}$ for all values of gate voltage from −3 V to 3 V were observed, and the device demonstrated ambipolar behavior under different temperatures (Fig. 3a). Under 500 °C, the WSe$_2$ FET still showed good transfer properties. Linear and symmetric $I_{ds}$-$V_{ds}$ curves obtained at 500 °C further demonstrated ambipolar behavior and suggested near-ohmic contact between WSe$_2$ and GF electrodes (Fig. 3b). Photoluminescence spectra illustrated that the bandgap of WSe$_2$ decreased from 1.55 eV to 1.40 eV as temperature increased from 150 °C to 500 °C (Supplementary Fig. 12). Theoretically, smaller bandgap resulted in smaller on/off ratio. As temperature varied from 20 °C to 500 °C, on/off ratio of the WSe$_2$ FET decreased from $2 \times 10^6$ to $1 \times 10^2$ (Fig. 3c). According to the

equation[30]:

$$SS = \ln 10 \cdot \frac{kT}{q} \cdot \frac{C_{ox} + C_s}{C_{ox}} \qquad (1)$$

where $k$ is Boltzmann constant, $T$ is absolute temperature, $q$ is the charge per carrier, $C_{ox}$ and $C_s$ are dielectric capacity and depletion capacity, respectively, higher temperature leads to larger SS. The SS of our device increased from 130 mV/dec to 600 mV/dec as the temperature varied from 20 °C to 500 °C (Fig. 3c). After 500 °C heating in air, the device was tested at room temperature again. Interestingly, due to the high temperature (500 °C) annealing, which improves WSe₂/GF contact, the electrical properties of our device did not degrade but slightly improved (SS became smaller, Supplementary Fig. 13). Lower resistance of GF electrode was observed after high temperature annealing (Supplementary Fig. 14), indicating that annealing will not affect the conducting properties of GF electrode.

The off-state current $I_{off}$ of WSe₂ FET increased from $10^{-12}$ A to $10^{-8}$ A as temperature varied from 20 °C to 500 °C (Fig. 3d). The h-BN gate dielectric leakage current ($I_{gs}$) of the same device measured at the same $V_{gs}$ and $V_{ds}$ under the same temperature was 1-2 orders of magnitude smaller than $I_{off}$ (Fig. 3d), indicating that the off-state current of WSe₂ FET at high temperature is dominated by the intrinsic turn-off characteristics of WSe₂ instead of h-BN leakage current. Therefore, h-BN is not only a good oxygen-resistant coating, but also an effective high-temperature dielectric layer. The remarkable high-temperature isolation properties of h-BN contribute to the high-performance of our WSe₂ devices.

## WSe₂ photodetector with negative photoconductivity

We next explored the photoelectric characteristics of the WSe₂ photodetectors. Under 25 W/m² white light illumination at 20 °C, the $I_{ds}$-$V_{gs}$ transfer curve moved upward for all values of gate bias from −3 V to 3 V

(Fig. 4b), indicating positive photoconductivity (PPC). Interestingly, under high temperature (400 °C), white light illumination resulted in a left shift of transfer curve (Fig. 4c). Larger shift was observed as light intensity increased. The N branch of transfer curve mainly demonstrated PPC, while P branch demonstrated negative photoconductivity (NPC). NPC phenomenon has been observed in low dimensional materials at room temperature[31–33]. As $V_{gs}$ was set at a constant value of 3 V, $I_{ds}$ increased under white light illumination at 400 °C, and the WSe₂ device act as a PPC photodetector (Fig. 4d). In contrast, as $V_{gs}$ was set at 0 V, $I_{ds}$ decreased under illumination at 400 °C, and the WSe₂ device act as an NPC photodetector (Fig. 4e). Therefore, high-temperature reconfigurable photodetector was realized which can switch between NPC and PPC photodetector under the same temperature by adjusting gate voltage $V_{gs}$. NPC and PPC photodetector are building blocks of photoelectric logic gate. Reconfigurability makes it a great advantage for our device to be applied to photoelectric logic gate.

To investigate the origin of the unconventional NPC phenomenon, control groups were prepared and tested under the same experimental condition: 1) WSe₂ with GF electrodes and mica encapsulation (two mica flakes with the thickness of 40–60 nm were placed on top and bottom of WSe₂), 2) Bare WSe₂ with GF electrodes (without h-BN encapsulation), 3) Bare WSe₂ with Pt electrodes (without h-BN encapsulation), 4) WSe₂ with Pt electrodes and h-BN encapsulation. The high-temperature measurements were done in a short time to minimize the oxidation of WSe₂. $V_{gs}$=0 V, or $V_{gs}$ was not applied. The first 3 types of devices all demonstrated PPC from room temperature to 400 °C (Supplementary Figs. 15–17). Only the last type of devices demonstrated NPC at 400 °C (Supplementary Fig. 18), indicating that h-BN is responsible for the NPC phenomenon.

At relatively low temperature, photoexcited electron-hole pairs are restricted in WSe₂ channel due to the excellent insulation of h-BN, and can be extracted by applying $V_{ds}$. As such, the current increases

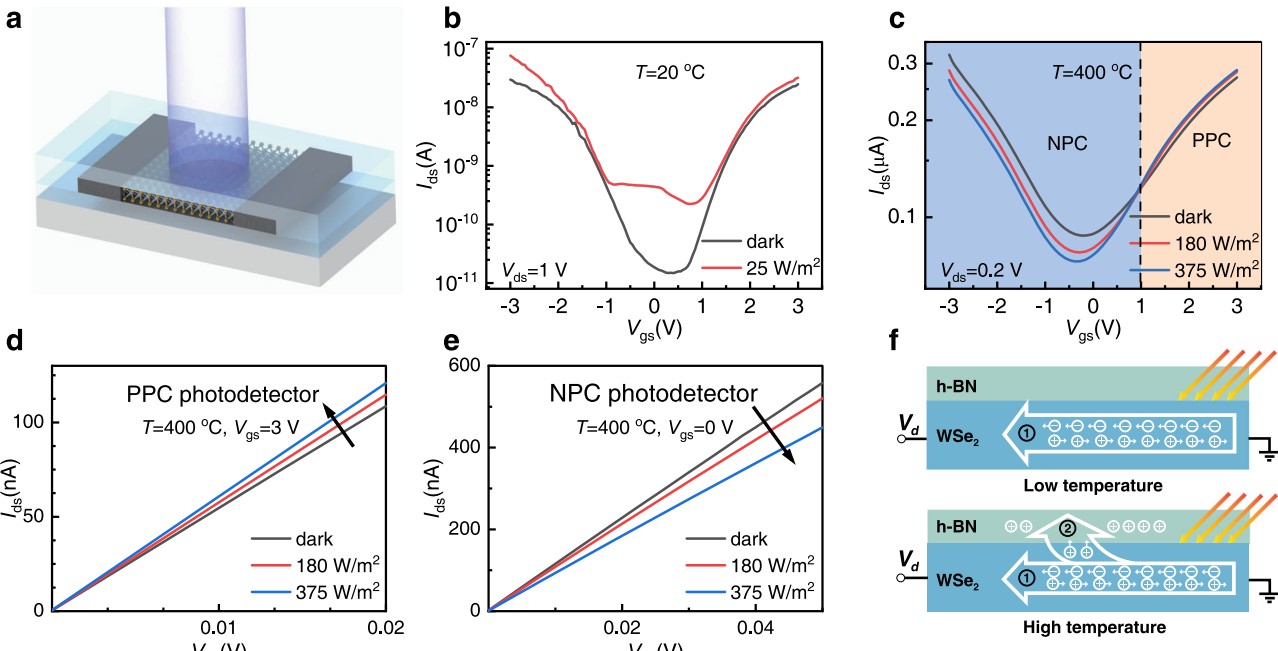

**Fig. 4 | WSe₂ photodetector with negative photoconductivity. a** Schematical image of WSe₂ photodetector under illumination. **b** Transfer curves of WSe₂ device under white light illumination at 20 °C. **c** Transfer curves of WSe₂ device under white light illumination at 400 °C. A white dashed line is used to distinguish the NPC device (blue area) and PPC device (orange area). **d** $I_{ds}$-$V_{ds}$ curves under white light illumination at 400 °C, $V_{gs}$ = 3 V. **e** $I_{ds}$-$V_{ds}$ curves under illumination at 400 °C,

$V_{gs}$=0 V. **f** Schematical views of optical sensing mechanism. At relatively low temperature, photoexcited electron-hole pairs are responsible for photocurrent (process 1). At high temperature, charge carriers have more chance to enter h-BN. Photogenerated holes trapped by the defect states in h-BN act as an equivalent positive gate bias (process 2).

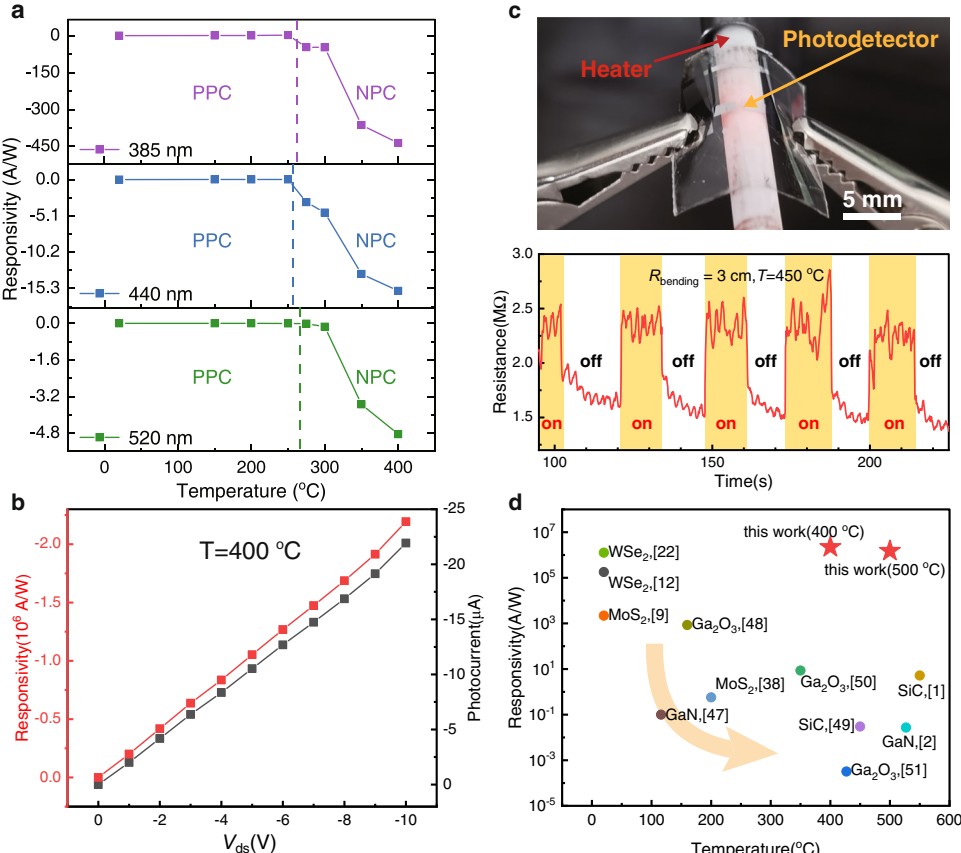

**Fig. 5 | High temperature optoelectronic performance. a** Responsivity of WSe$_2$ photodetector under 385 nm, 440 nm, and 520 nm wavelength illumination at different temperatures ($V_{gs}$ = 0). Purple (385 nm light), blue (440 nm light), green (520 nm light) dashed lines are used to indicate the switch temperature of PPC and NPC phenomenon. **b** Responsivity as a function of $V_{ds}$ derived from a typical WSe$_2$ photodetector at 400 °C. **c** High-temperature optical sensing under bending state ($V_{gs}$ = 0). A flexible photodetector is placed on a heating rod with temperature of 450 °C. Bending radius is 3 cm. The dynamic resistance variation demonstrates negative photoconductivity. The average rise time is ~0.3 s and the recovery time is about 13 s. **d** Comparison of our photodetector and state-of-the-art photodetectors.

after photoexcitation (process 1 in Fig. 4f). The h-BN gate dielectric leakage current increased from 10$^{-13}$ A to 10$^{-10}$ A as temperature varied from 20 °C to 500 °C (Fig. 3d), implying that the chance for charge carriers to enter h-BN layer significantly increases at elevated temperature. Photogenerated holes trapped by the defect states inside h-BN act as an equivalent positive gate voltage (photogating effect, process 2 in Fig. 4f), leading to a left shift of $I_{ds}$-$V_{gs}$ transfer curve as shown in Fig. 4c and Supplementary Fig. 19. Photogenerated charge carriers trapped by the defect state inside low-dimensional materials is one of the main reasons for NPC phenomenon[34]. For example, Xu et al., reported room temperature NPC phenomenon of graphene/black phosphorus heterojunction which is due to the electrons trapped in black phosphorus[35].

To figure out the temperature that process 2 starts to dominate and the impact of light wavelength, we utilized 385 nm wavelength purple light (18 W/m$^2$), 440 nm wavelength blue light (480 W/m$^2$), and 532 nm wavelength green light (440 W/m$^2$) to illuminate the WSe$_2$ photodetector respectively ($V_{gs}$ = 0 V) under different temperatures. The photodetector exhibited PPC at low temperatures (Fig. 5a). For all the three wavelengths, the device switched to NPC as temperature reached ~275 °C, indicating that NPC phenomenon happens in a wide wavelength range. It is worth mentioning that the NPC occurs due to the defect states inside h-BN, which means that different WSe$_2$ photodetectors may exhibit slightly different PPC/NPC transition temperatures. The absolute value of NPC responsivity was significantly larger than that of PPC at room temperature (under 385 nm illumination, the responsivity at 400 °C was ~2000-fold higher than that at

room temperature), and shorter wavelengths usually resulted in higher responsivity at the same temperature. Our devices exhibited ultrahigh responsivity of $2.2 \times 10^6$ A/W at 400 °C under 0.2 W/m$^2$ 365 nm illumination (Fig. 5b and Fig. S17). At 500 °C, we also obtained an impressive photoresponsivity of $1.1 \times 10^6$ A/W (Supplementary Fig. 20). The photoresponsivity is not only higher than that of state-of-the-art high-temperature photodetectors, but also higher than that of existing WSe$_2$ devices (Fig. 5d). High responsivity at high temperatures results in a large signal-to-noise ratio of 152.9, which is sufficient to meet the needs of optical sensing (Supplementary Fig. 21 and Supplementary Table 2). The detectivity D* can be derived from[12]:

$$D^* = \frac{A^{1/2}R}{(2qI_d)^{1/2}} \qquad (2)$$

where $A$ is the sensing area, $R$ is the photoresponsivity, $q$ is the unit of charge and $I_d$ is the dark drain current. Our device demonstrate detectivity of $1.6 \times 10^{13}$ Jones at 400 °C and $2.63 \times 10^{12}$ Jones at 500 °C. The detectivity of our devices is comparable to those of state-of-the-art 2D materials, III-V materials, and Si photodetectors (Supplementary Table 3).

## In situ high-temperature optical sensing under bending state

Flexible WSe$_2$ photodetector with a bending radius of 3 cm was attached on a ceramic heating rod with temperature of 450 °C. We utilized 18 W/m$^2$ 385 nm wavelength light to illuminate the device ($V_{gs}$ = 0 V). Its optical sensing performance is coupled with the impact

of strain (~0.17%) and high temperature. The dynamic resistance variation is shown in Fig. 5c. The resistance of the device increased rapidly from 1.5 MΩ to 2.5 MΩ within 0.3 s after illumination, indicating NPC phenomenon. As the light was turned off, it took approximately 13 s for the resistance to dropped back to the dark state value. This process was repeated for over 5 times and demonstrated good repeatability. Therefore, our flexible photodetectors can adapt to non-coplanar working conditions with good optical sensing performance which cannot be achieved by conventional rigid high-temperature photodetectors.

## Discussion

We developed $WSe_2$ photodetectors which can endure temperature up to 700 °C in air and 1000 °C in vacuum. Our research greatly expands the working temperature range of 2D materials, allowing the good properties of 2D materials to be applied in high temperature environments. The device exhibited unconventional NPC phenomenon, and the photoresponsivity in NPC mode ($2.2 \times 10^6$ A/W) is ~$10^5$-fold higher than that of existing high-temperature photodetectors, and also higher than that of existing $WSe_2$ photodetectors. This work bridges the technology gap between highly-sensitive photodetectors and high-temperature photodetectors. Current high-temperature photodetectors are rigid, which largely limits their applications. Our device is both highly flexible and can endure ultrahigh temperature and thus we realized in situ high-temperature optical sensing under bending state. Our approach opens up opportunities for 2D-material devices working in harsh environment, and may stimulate fundamental research of the fascinating properties and new phenomena of 2D materials at high temperature.

## Methods

### Device fabrication

A freshly cleaved mica substrate was firstly immersed in acetone, alcohol, and deionized water successively for 2 min ultrasonic cleaning in order to remove possible impurities. Then, bottom h-BN (40–60 nm), $WSe_2$ (10–20 nm), two GF electrodes and top h-BN (40–60 nm) were mechanically exfoliated using scotch tape and transferred onto mica substrate by PDMS film. The transfer process was carried out using an accurate transfer platform (Metatest, E1-T). Finally, a platinum electrode of 30 nm thick as top gate was fabricated using photolithography, metal deposition, and a lift-off process. As-fabricated devices were annealed in $N_2$ atmosphere at 250 °C for 30 min to remove residuals between the layers.

### Device characterizations

The thicknesses of the 2D materials and the surface roughness of GF and Pt were characterized by AFM (Bruker, Dimension Icon) using ScanAsyst-air mode. The Raman spectra were derived by Raman spectrometer (HORIBA Jobin Yvon, LabRAM HR Evolution) with 532 nm laser ($0.325 mW/cm^2$) and acquisition time of 120 s. The cross-sectional images of van der Waals heterostructures and h-BN/Pt interface were characterized by high-resolution transmission electron microscopy (JEOL-2100 TEM) with acceleration voltage of 200 kV. The samples for HRTEM were prepared using Xenon focused ion beam (Helios G4). High-temperature photoluminescence (PL) spectrum was detected on a temperature-control stage (Inspec HCP621G+) by PL spectrometer (Andor KYMERA-328I-B1, grating 300 grooves/mm), and a 532 nm continuous wave laser (commercial Coherent Genesis MX, intensity = 0.05 mW) was applied.

### Device measurements

The transfer characteristics and I-t characteristics of the $WSe_2$ FETs were measured by semiconductor parameter analyzer (Agilent B1500A). The devices were fixed and tested on a temperature-control stage. For the purpose of investigating electrical characteristics of the

devices under strain, we fixed it on the home-made matrix with different bending radius, and then tested it with semiconductor parameter analyzer. The dynamic resistance variation of the photodetectors was measurement by digital multimeter (Agilent 34470A). The devices were fixed and tested on a ceramic heating rod.

## Data availability

Relevant data supporting the key findings of this study are available within the article and the Supplementary Information file. All raw data generated during the current study are available from the corresponding authors upon request. Source data are provided with this paper.

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

## Acknowledgements

P. L. acknowledges financial support from National Natural Science Foundation of China (Grants No. 51775306) and Beijing Municipal Natural Science Foundation (Grants No. 4192027). The authors thank Prof. Rong Zhao for the discussion.

## Author contributions

P.L. conceived the experiments. P.L., Y.Z., and Z.Z. fabricated the devices and performed the electrical measurements. P.L., Y.Z., and J.Y. performed the photoelectrical measurements. L.L. and G.H. performed the PL measurements. P.L., Y.Z. and Y.T. analysed the results. P.L., Y.Z. and Y.T. co-wrote the manuscript. Z.Y. Supervised the project. All authors discussed the results and commented on the manuscript.

## Competing interests

The authors declare no competing interests.
