## [Peer Review File · Nature Communications]

High-Temperature Flexible WSe₂ Photodetectors with Ultrahigh PhotoresponsivityREVIEWER COMMENTS

Reviewer #1 (Remarks to the Author):

This manuscript reports high-temperature and high-photoresponsivity photodetectors based on flexible WSe₂. The temperature endurance of WSe₂ photodetectors is up to 700 °C in air and 1000 °C in a vacuum, respectively. In particular, the authors show detailed data to explain the negative photoconductivity behaviors of WSe₂ PDs at high temperature. The experiments were performed carefully and the results are quite interesting. In my opinion, the work is complete and warrants publication in Nature Communications provided the following points are addressed:

1. Authors mentioned that the graphene electrodes are good for high-temperature protection due to the suppression of oxygen molecule diffusion at good interfaces of graphene/h-BN. Could authors show the size of the top h-BN layer of the WSe₂ device with Pt electrodes compared with Fig. 1c? The size of the top h-BN layer of the WSe₂ device with Pt electrodes maybe affects the high-temperature protection results.
2. Authors showed the temperature-resistant capability (700 °C in air and 1000 °C in a vacuum) of WSe₂ devices via Raman spectrum (Fig 2 and Fig S7). Could authors compare the dark current or I_{ds} - V_{ds} curves of WSe₂ devices with/without high-temperature annealing (700 °C in air and 1000 °C in a vacuum)?
3. In recent years, negative photoconductivity (NPC) in low-dimensional materials is reported a lot. I suggest the authors should cite more papers in line 216 and 224 of Page 11. In particular, the authors mentioned that the NPC phenomenon in WSe₂ photodetectors results from photogenerated holes trapped by the defect states inside h-BN (line 242 of Page 12). I suggest the authors should cite more papers to emphasize this mechanism.
4. Authors mentioned that their devices exhibit ultrahigh responsivity of 2.2×10^6 A/W at 400 °C and 1.1×10^6 A/W at 500 °C under 0.2 W/m² 365 nm illumination (Fig. 5b and Fig. S17). The responsivity of WSe₂ photodetectors decreases with increasing annealing temperature. Is the negative photoconductivity (NPC) behavior in WSe₂ photodetectors similar to responsivity as the temperature increase (> 400 °C)?
5. Finally, while photoresponsivity and the response time are important figures of merit, photodetectivity is another critical parameter for a photodetector that is not mentioned here. It is important to report on the photodetectivity of the WSe₂ photodetectors. The authors should calculate/measure this number and then compare their devices to other photodetectors based on 2D materials, state-of the art III-V or Si devices.

Reviewer #2 (Remarks to the Author):

The authors reported a flexible h-BN-encapsulated WSe₂ photodetector with high-temperature tolerance. They use the h-BN-encapsulation method to protect WSe₂ from oxidation at high temperatures, a method that has been widely used in previous studies on two-dimensional (2D) materials such as graphene and transition metal dichalcogenides (TMDs). In this manuscript, the authors achieved impressive results, their photodetectors can work with higher performance of photoresponsivity at over 700°C. The data and analysis are quite comprehensive, and the supplementary materials provide answers to most of the questions I have in mind, except for the following:

1. The carriers are excited at high temperatures, and the current is more than three orders of magnitude higher than that at room temperature. This may indeed yield large values of responsivity when utilizing the formula for responsivity at high temperatures ($\sim 10^6$ A/W at 400 °C). But the high temperature will also greatly amplify the noise, for the sensors/detectors, the signal-to-noise ratio is often more important. How do these

devices perform in terms of the signal-to-noise ratio at different temperatures?

2. In Figure 4c, the current reaches over 0.3 A and the power is close to 1 W. This is a very large power for a device of just about 100 square microns (the power density >105 W/cm²), how does the WSe₂ withstand such energy?

3. Considering the structure of the device (fig 1d), where the thickness of the "graphene electrode" is more than 15 nm and the thickness of WSe₂ is more than 10 nm, strictly speaking, these structures are no longer 2D, it is more appropriate to use "graphite flake" than "graphene".

Technical issues

1. The first 2D should be "two-dimensional (2D)".

2. Page 2:"and even higher than that of existing WSe₂ photodetectors." should be " also higher than 1.8×10^5 A/W of existing WSe₂ photodetectors."

The manuscript should be acceptable after minor revision.

Response to Reviewer #1

This manuscript reports high-temperature and high-photoresponsivity photodetectors based on flexible WSe₂. The temperature endurance of WSe₂ photodetectors is up to 700 °C in air and 1000 °C in a vacuum, respectively. In particular, the authors show detailed data to explain the negative photoconductivity behaviors of WSe₂ PDs at high temperature. The experiments were performed carefully and the results are quite interesting. In my opinion, the work is complete and warrants publication in Nature Communications provided the following points are addressed:

1. Authors mentioned that the graphene electrodes are good for high-temperature protection due to the suppression of oxygen molecule diffusion at good interfaces of graphene/h-BN. Could authors show the size of the top h-BN layer of the WSe₂ device with Pt electrodes compared with Fig. 1c? The size of the top h-BN layer of the WSe₂ device with Pt electrodes maybe affects the high-temperature protection results.

Response: This comment is helpful to our manuscript. The zoom out images below show that the top h-BN in Fig. 2d is approximately 150 μm × 150 μm, which is larger than the top h-BN shown in Fig. 1c.

Figure S18. Zoom out image of the Pt-electrode device shown in Fig. 2d, e.

We have tried top h-BN flakes with various sizes, and the results are listed below (Table S1). Pt-electrode devices were all oxidized after annealing, while graphite-flake-electrode devices all demonstrated negligible change. Figure S22a, b show a graphite-flake-electrode devices with ~15000 μm² h-BN which demonstrated negligible change after annealing. Figure S22c, d show a Pt-electrode devices with ~15000 μm² h-BN which demonstrated strong oxidation after annealing. The size of the top h-BN layer has negligible effect on the high temperature protection results.

Table S1. Devices with different sizes of top h-BN after annealing

Size of top h-BN	Pt electrode devices	Graphite flake electrode devices
~5000 μm^2	oxidized	unchanged
~10000 μm^2	oxidized	unchanged
~15000 μm^2	oxidized	unchanged
~18000 μm^2	oxidized	unchanged
~22500 μm^2	oxidized	/

Figure 1. (a, b) Graphite-flake-electrode WSe₂ device with ~15000 μm^2 top h-BN before and after annealing. (c, d) Pt-electrode WSe₂ device with ~15000 μm^2 top h-BN before and after annealing.

2. Authors showed the temperature-resistant capability (700 °C in air and 1000 °C in a vacuum) of WSe₂ devices via Raman spectrum (Fig 2 and Fig S7). Could authors compare the dark current or I_{ds} - V_{ds} curves of WSe₂ devices with/without high-temperature annealing (700 °C in air and 1000 °C in a vacuum)?

Response: This comment is helpful to our manuscript. The I_{ds} - V_{ds} curve before and after 700 °C in air and 1000 °C in vacuum annealing are shown below. In both situations, the current increased after high temperature annealing. We assume two factors contribute to this phenomenon: 1) Better contact between WSe₂ and graphite flake. 2) The mismatch of the thermal expansion coefficient

between 2D materials and substrate results in built-in-strain after annealing. The electrical properties of WSe₂ are very sensitive to strain.

Figure S19. I_{ds} - V_{ds} curve before and after 700 °C in air and 1000 °C in vacuum annealing.

3. In recent years, negative photoconductivity (NPC) in low-dimensional materials is reported a lot. I suggest the authors should cite more papers in line 216 and 224 of Page 11. In particular, the authors mentioned that the NPC phenomenon in WSe₂ photodetectors results from photogenerated holes trapped by the defect states inside h-BN (line 242 of Page 12). I suggest the authors should cite more papers to emphasize this mechanism.

Response: This comment is helpful to our manuscript. We have cited papers about NPC in low-dimensional materials in revised manuscript (ref. 47-51).

Photogenerated charge carriers trapped by the defect state inside low-dimensional materials is one of the main reasons for NPC phenomenon (at room temperature). For example, Xu et al., reported room temperature NPC phenomenon of graphene/black phosphorus heterojunction which is due to the electrons trapped in black phosphorus (ref. 51 in revised manuscript).

4. Authors mentioned that their devices exhibit ultrahigh responsivity of 2.2×10^6 A/W at 400 °C and 1.1×10^6 A/W at 500 °C under 0.2 W/m^2 365 nm illumination (Fig. 5b and Fig. S17). The responsivity of WSe₂ photodetectors decreases with increasing annealing temperature. Is the negative photoconductivity (NPC) behavior in WSe₂ photodetectors similar to responsivity as the temperature increase (> 400 °C)?

Response: The responsivities of 2.2×10^6 A/W at 400 °C and 1.1×10^6 A/W at 500 °C were all derived from NPC mode.

At higher temperature, more holes are trapped inside h-BN after illumination, resulting in larger left shift of transfer curve, as shown in the schematic below. The red line is the dark current, and the blue line is the current under illumination. Due to the ambipolar behavior of WSe₂, the NPC photocurrent ΔI at $V_g = V_{test}$ increases as temperature increases from T_1 to T_2 , but decreases as temperature increases from T_2 to T_3 . Therefore, whether NPC photoresponsivity increases as temperature increases depends on temperature and also gate voltage V_{test} .

Figure S20. Schematic of transfer curves at different temperatures.

5. Finally, while photoresponsivity and the response time are important figures of merit, photodetectivity is another critical parameter for a photodetector that is not mentioned here. It is important to report on the photodetectivity of the WSe₂ photodetectors. The authors should calculate/measure this number and then compare their devices to other photodetectors based on 2D materials, state-of-the-art III-V or Si devices.

Response: This comment is helpful to our manuscript. The detectivity can be derived from (Zhang, W. et al. *ACS nano* **8**, 8653-8661 (2014)):

$$D^* = \frac{A^{1/2}R}{(2qI_d)^{1/2}}$$

Where A is the sensing area, R is the photoresponsivity, q is the unit of charge, and I_d is the dark current. Our device demonstrates detectivity of 1.6×10^{13} Jones at 400°C and 2.63×10^{12} Jones at 500°C.

The comparison of representative photodetectors is shown in Table S2 in revised Supplementary Information. The detectivity of our devices is comparable to those of state-of-the-art 2D materials, III-V materials, and Si photodetectors.

Table S2. Comparison of representative photodetectors

Materials	Wavelength[nm]	Temperature[°C]	D*[jones]	Reference
WSe ₂ (CVD)	500-900	RT	1×10 ¹⁴	[1]
MoS ₂	532	200	1×10 ¹⁰	[2]
MoS ₂	550-800	RT	7.7×10 ¹¹	[3]
WSe ₂	532	RT	1.1×10 ¹²	[4]
GaN	440	527	4×10 ⁸	[5]
Ga ₂ O ₃	270	RT	1×10 ¹²	[6]
Al _{0.4} Ga _{0.6} N	280	127	2.4×10 ¹³	[7]
SiC nanowire	254	RT	7.2×10 ¹⁰	[8]
Si	1060	RT	1×10 ¹⁰	[9]
Si	950	RT	1.5×10 ¹⁴	[10]
WSe ₂	365	400	1.6×10 ¹³	This work
WSe ₂	365	500	2.63×10 ¹²	This work

Response to Reviewer #2

Recommendation: The manuscript should be acceptable after minor revision.

The authors reported a flexible h-BN-encapsulated WSe₂ photodetector with high-temperature tolerance. They use the h-BN-encapsulation method to protect WSe₂ from oxidation at high temperatures, a method that has been widely used in previous studies on two-dimensional (2D) materials such as graphene and transition metal dichalcogenides (TMDs). In this manuscript, the authors achieved impressive results, their photodetectors can work with higher performance of photoresponsivity at over 700 °C . The data and analysis are quite comprehensive, and the supplementary materials provide answers to most of the questions I have in mind, except for the following:

1. The carriers are excited at high temperatures, and the current is more than three orders of magnitude higher than that at room temperature. This may indeed yield large values of responsivity when utilizing the formula for responsivity at high temperatures ($\sim 10^6$ A/W at 400 °C). But the high temperature will also greatly amplify the noise, for the sensors/detectors, the signal-to-noise ratio is often more important. How do these devices perform in terms of the signal-to-noise ratio at different temperatures?

Response: This comment is helpful to our manuscript. We measured the noise of the device at 20°C, 200°C, and 400°C, respectively. Fig. S21 shows noise current sequence at different temperatures.

Figure S21. Noise current sequence at different temperatures.

The noise currents, signal currents and signal-to-noise ratio at different temperatures are listed in Table S1. We use standard deviation of noise to represent I_{noise} . The I_{signal} values were all derived under 18 W/m^2 385 nm illumination. Compared to room temperature, the signal-to-noise ratio at 200°C is significantly lower. This is because the high temperature greatly amplifies the noise while the PPC signal does not significantly increase. At 400°C , the device operated in NPC mode has a very high responsivity, resulting in an increased signal-to-noise ratio (152.9). The signal-to-noise ratio of our device at high temperature is sufficient to meet the needs of optical sensing.

Table S3. Signal-to-noise ratio of the device at different temperatures

Temperature($^\circ\text{C}$)	$I_{noise}(\text{A})$	$I_{signal}(\text{A})$	SNR	Device status
20	2.74×10^{-12}	3.11×10^{-9}	1135	PPC
200	4.64×10^{-10}	1.31×10^{-8}	28.2	PPC
400	2.06×10^{-9}	3.15×10^{-7}	152.9	NPC

2. In Figure 4c, the current reaches over 0.3 A and the power is close to 1 W. This is a very large power for a device of just about 100 square microns (the power density $>10^5 \text{ W/cm}^2$), how does the WSe_2 withstand such energy?

Response: We thank the reviewer for pointing this out. The unit “A” should be “ μA ” in Figure 4c. We have modified the figure in revised manuscript.

3. Considering the structure of the device (fig 1d), where the thickness of the "graphene electrode" is more than 15 nm and the thickness of WSe_2 is more than 10 nm, strictly speaking, these structures are no longer 2D, it is more appropriate to use "graphite flake" than "graphene".

Response: This comment is helpful to our manuscript. We used "graphite flake (GF)" instead of "graphene" in revised manuscript.

Technical issues

1. The first 2D should be “two-dimensional (2D)”.

Response: We thank the reviewer for pointing this out. We have modified it in revised manuscript.

2. Page 2:“and even higher than that of existing WSe_2 photodetectors.” should be “ also higher than 1.8×10^5 A/W of existing WSe_2 photodetectors.”

Response: This comment is helpful to our manuscript. We have changed “and even higher than that of existing WSe_2 photodetectors.” into “also higher than 1.8×10^5 A/W of existing WSe_2 photodetectors.” in revised manuscript.

REVIEWERS' COMMENTS

Reviewer #1 (Remarks to the Author):

I have no further comments. Thanks for the revisions

Reviewer #2 (Remarks to the Author):

My concerns have been fully addressed. I would recommend this revised version to be published in Nature Communications.